# Improvement of Color and Oxidative Stabilities in Nellore Bull Dark Meat in High-Oxygen Package by Lactate and Rosemary Oil Extract

**DOI:** 10.3390/foods12061302

**Published:** 2023-03-18

**Authors:** Caio César de Sousa Ribeiro, Kathelyn Araújo Guimarães, Eduardo Francisquine Delgado, Júlio César de Carvalho Balieiro, Anna Cecilia Venturini, Carmen Josefina Contreras Castillo

**Affiliations:** 1Agri-Food Industry, Food and Nutrition, Luiz de Queiroz College of Agriculture, University of Sao Paulo, Piracicaba 13418-900, Brazil; 2Núcleo de Apoio Técnico ao Ensino, Pesquisa e Extensão, Institute of Environmental, Chemical and Pharmaceutical Sciences, Federal University of Sao Paulo, Diadema 09972-270, Brazil; 3Department of Animal Science, Luiz de Queiroz College of Agriculture, University of Sao Paulo, Piracicaba 13418-900, Brazil; 4Department of Nutrition and Animal Production, School of Veterinary Medicine and Animal Science, University of Sao Paulo, Sao Paulo 13635-900, Brazil; 5Department of Pharmaceutical Sciences, Institute of Environmental, Chemical and Pharmaceutical Sciences, Federal University of Sao Paulo, Diadema 09913-030, Brazil

**Keywords:** ultimate pH, oxygen consumption, metmyoglobin-reducing activity, TBARS

## Abstract

This study aimed to improve the color and oxidative stabilities of dark Nellore bull steaks with greater-than-normal ultimate pH (pH_u_) by the injection (8% raw wet weight basis) of a solution with L-lactate (2.5%), phosphate (0.3%) and rosemary extract (0.06%), with further packaging in high oxygen atmosphere (HiOx MAP). *Longissimus lumborum* muscles from pasture-fed Nellore bulls were divided into three pH_u_ ranges: normal (<5.80), intermediate (5.81–6.19), and high (≥6.2). Muscles were then halved, with sections were randomly assigned to non-enhanced (C, *n* = 6/pH_u_ range) or injected (E, *n* = 6/pH_u_ range) groups, at 72 h postmortem. Each section was cut into 2 cm-slices, which were HiOx-packed and then stored for 5 days (dark) and displayed for 9 days (fluorescent lighting) at 2 °C. Higher pH_u_ steaks exhibited greater a*, b*, h*, C* and surface oxymyoglobin and lower surface deoxymyoglobin and oxygen consumption compared to those of normal pH_u_ between days 0 and 5 (*p* < 0.05). Over the time, normal-pH_u_ muscles showed oxidative protection (lower TBARS and greater metmyoglobin reducing ability values, *p* < 0.05) in enhanced-steaks. Therefore, enhancement and HiOx MAP seem to produce greater-than-normal pH_u_ Nellore bull steaks with a preferable color and quality, even after display time.

## 1. Introduction

Brazil slaughtered 39.1 million bovine heads in 2021 [1], with predominant (~80%) zebu genetics (*Bos taurus indicus*, mainly Nellore breed) [2]. Nearly 83% of the national herd was raised under extensive grazing conditions [1], which provide a diet rich in polyunsaturated fatty acids (PUFAs) and antioxidants [3,4].

The combination of *Bos taurus indicus* bulls’ irritable behavior [5], greater muscle oxidative efficiency and pH drop resistance [6], and grass finishing diet may result in lower muscle glycogen content and increased oxidative metabolism [7]. These factors contribute to limiting the postmortem pH drop to a normal range of 5.4–5.8 and then producing a greater-than-normal pH_u_ (ultimate pH), which affects meat color development [8,9].

The problem with greater-than-normal pH_u_ has been reported, where Nellore bulls slaughtered in a Brazilian commercial plant had 53.1% of the 399 *Longissimus lumborum* (LL) muscles sampled presented pH_u_ > 5.8 (24 h postmortem), while 5.5% of those had pH > 6.3 [10]. Darker meat due to increased pH_u_ is related to increased water-holding capacity (WHC) of myofibrillar proteins, with decreased light scattering inside the structure of fiber [11].

Dark cuts have concerned meat processors as color is an initial attribute that influences beef purchasing. Particularly, the importance of appearance aspects (overall, freshness, and red color) to the beef market was demonstrated by Boito et al. [12], who detailed that purchasers demand bright-cherry red steaks, albeit beef color is not always related to eating acceptability [13]. Therefore, any variation in the desired color may negatively impact beef marketability.

Recent studies have demonstrated that pH-related darkening can be solved by packaging steaks in a modified atmosphere with an oxygen concentration (HiOx-MAP) greater than 50% [14,15]. However, high partial pressure of oxygen, especially combined with fluorescent light exposure, and a higher enzymatic activity of Δ9-desaturase in Nellore beef [16] can stimulate lipid, protein, and myoglobin (Mb) oxidation [14,17,18] and then deteriorate meat quality.

On the other hand, oxidation can be inhibited by adding antioxidants/antimicrobials, such as phosphates, plant extracts, such as rosemary extract, ascorbic acid, and glycolytic and tricarboxylic intermediates (lactate, malate, pyruvate, succinate) [19,20,21]. Enhancement with lactate and rosemary extract, especially due to phenolic diterpenes, carnosic acid, and carnosol, has been shown to improve and stabilize meat color, besides their preservative effects that contribute to extending meat shelf-life [21,22,23].

Given the molecular specificities from Nellore genetics [24,25] and the management system widely practiced in several meat-producing countries, the present study aimed to determine the effect of meat enhancement with lactate, phosphate, and rosemary extract in Nellore’s dark-cutting-beef packaged in HiOx-MAP (80% O_2_/20% CO_2_) in order to limit time-related lipid and Mb oxidation.

## 2. Materials and Methods

### 2.1. Raw Materials, Processing, and Store/Display Conditions

Eighteen pasture-raised, 30–36 months old Nellore bulls were harvested at a commercial slaughterhouse in Barretos, SP, Brazil, chilled at 4 °C for 48 h, and then sorted into grouped into the following three pH_u_ ranges: normal (N—5.40–5.79), intermediate (I—5.80–6.19), and high (H—pH_u_ ≥ 6.20), according to the pH measured at three different points of LL muscle between the 1st and 6th vertebra with a penetration pH meter (1140 model, Mettler Toledo, Urdorf, Switzerland).

The selected carcasses were then fabricated into primal cuts, with LL muscles identification, vacuum packaging (VSA 211, Cryovac, Sealed Air, New Jersey, NJ, USA), and transport to the laboratory where samples were kept at 2 °C until 72 h postmortem. After superficial fat removal, each loin was divided into two equal sections, resulting in 36 half muscles, which were randomly assigned to non-enhanced or control (C group, *n* = 6/pH_u_ range) and lactate-phosphate-rosemary-enhanced (E group, *n* = 6/pH_u_ range).

Enhanced sections underwent injection of a brine solution (pH 6.4), whose composition is present in Table 1, with a multi-needle injector (Super Inject Max Power Flavor, model Stander) up to approximately 8% of raw wet weight basis. The average injection levels for each pH_u_ range were calculated 30 min after pumping by the following formula: [(weight after injection − raw weight)]/raw weight]/100, as well as the average concentration of each compound per pH range (Table 1).

All muscle sections were then sliced into steaks (2.0 cm) and placed in preformed trays with soaker pads (trays model 13D65, 24 cm × 16 cm × 6.5 cm, white polypropylene with ethylene vinyl alcohol (EVOH) barrier, O_2_ permeability < 0.5 cm^3^/m^2^/24 h at 50% relative humidity, Sealed Air Corp., New Jersey, USA Air was removed from each tray, and a high oxygen atmosphere (HiOx-MAP) (80% O_2_/20% CO_2_; Air Liquide Ltd., Campinas, Brazil) was flushed, with subsequent sealing with a high barrier sealing film (4532-G lid, with nominal thickness of 70 µm, O_2_ permeability < 5 cm^3^/m^2^/24 h at 23 °C, 66% of relative humidity and water vapor permeability < 5 g/m^2^/24 h at 38 °C e 90% relative humidity, Bemis Company—Dixie Toga, Brazil) by using a Multivac machine (model T200, Multivac, Ltd., Wolfertschwenden, Germany).

High-oxygen packed steaks were stored in a cold room at 2 ± 1 °C for 5 days, when steaks began to be exposed to fluorescent light (1980 ± 150 lux, MLM-1010, Minipa, Sao Paulo, Brazil) for additional 9 days (Figure 1). The cold room was defrosted at 4-h intervals, and the trays were rotated once a day to ensure a minimal difference in the fluorescent light and temperature exposures among the steaks. During MAP time, steaks were removed from the cold chamber and underwent the analyses, as shown in Figure 1.

### 2.2. Headspace Composition and pH

The gas composition (%CO_2_ and %O_2_, *v*/*v*) was monitored by headspace oxygen/carbon dioxide analyzer (CheckPoint II, PBI Dansensor A/S, Ringsted, Denmark). Gas measurements were performed by inserting the needle through a rubber septum attached to the top of the tray.

The pH was measured immediately after the tray cover removal, at three random points in each steak, with a calibrated (pH 4.0, 7.0, 10.0) pH meter (pH 1140 model, Mettler Toledo, Urdorf, Switzerland) with automatic temperature compensation.

### 2.3. Instrumental Color

On the surface of each steak, CIE lightness (L*), redness (a*), and yellowness (b*) were measured using a Hunter MiniScan XE Plus spectrophotometer (Hunter-Lab, Reston, VA, USA), integrated into an Easy Match QC system, with 2.54 cm diameter aperture, 10° standard observer and illuminant A. Readings were performed at five random locations on steak surfaces. The spectrophotometer was calibrated using black-and-white reference standards provided by the manufacturer. From a* and b* values, color stability was assessed by the calculation of Hue angle (arctan (b*/a*) and a*/b*, whereas color intensity was assessed by chroma (a*2 + b*2)½.

### 2.4. Myoglobin Redox States, Nitric Oxide Metmyoglobin-Reducing Ability and Oxygen Consumption

The proportions of the surface Mb redox states were estimated using the ratio of the absorption and scattering coefficients (K/S) forms of specific reflectances (R) from the Hunter readings, which were transformed into K/S forms by the following formula: (1 − R)2/(2R). Myoglobin (Mb) estimations were calculated by following equations: Deoxymyoglogin (DeoxyMb, K/S473/K/S525), Metmyoglobin (MetMb, K/S_572_/K/S_525_), Oxymyoglobin (OxyMb, K/S_610_/K/S_525_) [26]. As Hunter spectrophotometer only records reflectance values between 400 and 700 nm at 10 nm intervals, linear interpolation was used to calculate intermediate wavelength values. A smaller ratio indicates increase in Mb form proportion. In addition, such ratios were used to estimate the MetMb reducing ability (MRA) via Mb resistance to oxidation in nitric oxide (NO-Met), and the oxygen consumption (OC) via Mb resistance to oxygenize, as performed by McKeith et al. [8], due to the difficulty to obtain 100% of MetMb and OxyMb in greater-than-normal pH_u_ steaks.

A sample with 3 cm × 3 cm was cut from a free-fat and connective tissue area of each steak and divided into two halves horizontally, each one with 1 cm of height. The top half, already exposed to HiOx-MAP and light, was assigned to MRA method and then submerged in 50 mL of 0.3% (w/v) solution of sodium nitrite (Sigma Aldrich, MO, USA) for 20 min at 25 °C to oxidize Mb into NO-MetMb. Each sample was then removed from the solution, carefully dried, vacuum packaged (Sealed Air, New Jersey, NJ, USA), and immediately scanned in duplicate with a HunterLab MiniScan XE Plus spectrophotometer. A numerical lower ratio represents a lower MRA.

The bottom half of the sample (3 cm × 3 cm × 1 cm), whose upper surface comprised the interior of the steak, was assigned to OC method and allowed to oxygenate (bloom) for two hours at 2 ± 1 °C, with cover with oxygen-permeable film to prevent surface dehydration. After blooming, the sample was vacuum packaged (Sealed Air, Saddle Brook, NJ, USA) and immediately scanned with a Hunter Lab MiniScan XE Plus spectrophotometer. Like MRA method, a lower ratio represents a lower OC.

### 2.5. Lipid Oxidation

Malondialdehyde (MDA) was extracted from minced steaks in duplicate, as performed by Selani et al. [27]. For quantification, the absorbance of the extracts was measured with a UV–Vis mini 1240 spectrophotometer (Shimadzu, Kyoto, Japan) at 532 and 600 nm against a blank of ultrapure water. A standard curve was prepared with 1,1,3,3-tetraethoxypropane (from 0 to 0.4 nmol/L), and the thiobarbituric acid reactive substances (TBARS) content was expressed as mg of MDA/kg of sample.

### 2.6. Experimental Design and Statistical Analysis

The study followed a completely randomized design, with cross-classification scheme and repeated measures. A half LL muscle was considered for analysis and referred to different bulls (*n* = 6). Data were analyzed using a linear mixed model (LMM) by the SAS^®^ 9.4 software, using PROC MIXED procedure (SAS, Inst. Inc., Cary, NC, USA). For all variables, the mixed model included fixed factors Enhancement (C and E groups), pH_u_ ranges (N, I, and H), Time (0, 5, 8, 11, and 14 days or 0, 5, and 14 days for TBARS, MRA, and MetMb), Enhancement × pH_u_ ranges, Enhancement × Time, pH_u_ ranges × Time and Enhancement × pH_u_ ranges × Time interactions and samples nested within muscle as random effect. Least significant difference (LSD) was used for pairwise comparisons of the least square means by PDIFF option.

## 3. Results

### 3.1. Muscle pH and Headspace Gas Composition

The muscle pH values ranged between 5.55 and 6.94 at 48 h postmortem. In contrast to enhancement, pH_u_ ranges and time affected the muscle pH values and headspace gas composition, with interactions between these factors for pH (*p* < 0.0001) and the O_2_ (*p* = 0.0014) and CO_2_ (*p* < 0.0001) contents (Table 2).

The pH values measured at each day showed the pattern N < I < H (*p* < 0.01), which was maintained until the end of the experiment. Regarding time-related changes in each pH_u_ range, pH values varied less as pH_u_ increased. To illustrate such behavior, N samples jumped from 5.47 (day 0) to 5.82 (day 14) (variation of 0.35 units, *p* < 0.01), whereas I steaks rose from 5.83 (day 0) to 6.00 (day 14) (variation of 0.17 units, *p* < 0.01), and H steaks slightly varied from 6.28 (day 0) to 6.41 (day 14) (variation of 0.13 units, *p* > 0.05). 

The HiOx condition (80% O_2_/20% CO_2_) established at day 0 in the present study was preserved until day 14, despite variations over the time. Despite no difference among the three pH_u_ groups at day 0, at day 14, both gases showed distinction between N-I and H steaks, with those of H range presenting lower O_2_ (*p* < 0.01) and greater CO_2_ concentrations (*p* < 0.01).

Normal and intermediate pH_u_ steaks showed a significant reduction in the O_2_ concentration between days 5 and 8 (*p* < 0.001) and similar results after day 8 (*p* > 0.05). High pH_u_ steaks, however, decreased between days 0 and 5 (*p* < 0.01) and between days 5 and 14 (*p* < 0.01). Regarding the concentration of CO_2_, despite the increase between days 0 and 5 (*p* < 0.01), N steaks had similar CO_2_ content between days 0 and 14; whereas in the I steaks the this only changed, although gradually, between days 0 and 8 (*p* < 0.01). The samples in H range nevertheless showed a consistent increase in the CO_2_ proportion after storage (days 0–5, *p* < 0.01) and at the beginning of the display time (days 5–8, *p* < 0.01).

### 3.2. Surface Redness

Enhancement with lactate, phosphate, and rosemary increased the mitochondrial OC (C: 0.22 vs. E 0.26, *p* < 0.0001), which may have contributed to developing less lightness (lower L*, *p* < 0.0001), greater surface DeoxyMb, *p* = 0.0003), less redness (lower a*, *p* < 0.0001), lower hue angle (*p* = 0.0144), lower surface OxyMb (*p* < 0.0001), less yellowness (lower b*, *p* < 0.0001), and less saturated red color (lower chroma, *p* < 0.0001) on the surface of E steaks compared to those of C group (Table 3).

Lightness (L*) values, corresponding to visual, were affected by enhancement (*p* < 0.0001), pH_u_ range (*p* < 0.0001), and enhancement × pH_u_ ranges interactions (*p* < 0.0001). The normal and intermediate pH_u_ C steaks had greater L* values (48.64 and 47.78, respectively, SE: 0.48, *n* = 6, *p* < 0.01) than those of H range (43.11, SE: 0.48, *n* = 6), while no difference was observed among pH_u_ ranges within the E group (N: 43.99, I: 43.24, and H: 42.63, SE: 0.48, *n* = 6, *p* > 0.01). Compared to the C group, N and I steaks in the E group had lower L* values (darkening) (*p* < 0.01) but without differences between the groups in the H range samples.

The interaction pH_u_ range × the time in HiOx was significant (*p* < 0.05) for redness (a*), yellowness (b*), hue angle, chroma, surface DeoxyMb and OxyMb (Table 4), and OC (Figure 2), but no triple interaction was observed.

Before HiOx-MAP (day 0), steaks showed a related increase between OC and pH_u_, thereby forming an OC pattern at H > I > N ranges (*p* < 0.01). This distinction among pH_u_ ranges was also observed in the surface Mb redox states, with greater DeoxyMb (*p* < 0.05) and lower OxyMb (*p* < 0.05) as pH_u_ increased. Except for the hue angle values, which showed similarity between the N and I ranges, a*, b*, and chroma values declined as pH_u_ increased (N > I > H, *p* < 0.05).

After five days in HiOx-MAP, steaks in I and H ranges showed a prominent improvement in surface color, which resulted in statistical similarity (*p* > 0.05) among all three pH_u_ ranges for a*, b*, chroma, surface OxyMb, and OC.

Between days 0 and 5, I and H steaks exhibited a remarkable decline in OC (*p* < 0.01, Figure 2), which, however, was less observed in those E group steaks (C-I: 0.19 vs. E-I: 0.24 and C-H: 0.17 vs. E-H: 0.21, *p* < 0.01). Such reduction in OC was accompanied by an increase in surface OxyMb proportion (*p* < 0.05) and a decrease (*p* < 0.05) in surface DeoxyMb proportion.

Between days 5 and 8, N steaks exhibited impaired surface color by the decrease in a*, chroma, and surface OxyMb values (*p* < 0.05) and an increase (*p* < 0.05) in hue angle values. No change, however, was observed in I and H samples for these parameters until the end of display time.

As a consequence of changes throughout time, N steaks finished the experiment with a redder (a* and OxyMb) and more vivid (chroma) surface color in comparison to those in the I and H ranges (*p* < 0.05). On the other hand, hue angle and DeoxyMb results showed similarities between the N and I ranges, which were greater than those samples in the H range.

### 3.3. Oxidative Stability (Surface MetMb, MRA, and TBARS)

The results of MRA (*p* = 0.0037) and TBARS (*p* = 0.0130) showed a significant enhancement × pH_u_ range × time interaction (Table 5), whereas the surface MetMb proportion only resulted in the interaction between pH_u_ ranges and time (*p* < 0.0001) (Table 4).

Prior to HiOx-MAP (day 0), as pH_u_ increased, more MetMb proportion was present on steak surfaces (*p* < 0.05, Table 4). On the other hand, less NO-MetMb was formed in MRA analysis on H steaks either in C or E groups than in N and I ranges (*p* < 0.05), considering that enhancement did not affect MRA on day 0 within any pH_u_ range. This pH-dependent oxidative inhibition was also observed in the TBARS results in the C steaks for either the I or H range, with opposite results for N steaks (*p* < 0.05).

After 5 days under dark storage, MRA and lipid oxidation were affected by muscle pH_u_ (*p* < 0.05) and, mostly, by enhancement (*p* < 0.05). The enhanced group had greater MRA than the C group within I and H pH_u_ (*p* < 0.05). TBARS was lower for the E group compared to C group in the I steaks (*p* < 0.05).

At the end of display time (day 14), there was a noticeable impact of pH_u_ range on MRA, surface MetMb and lipid oxidation. Due to a lesser time-related accumulation of MetMb on the surface of the I and H steaks, these samples concluded the experiment was less brownish (*p* < 0.05) than those of the N range.

Enhanced or control steaks with high pH_u_ ended display time with greater MRA (*p* < 0.05) and lower TBARS (*p* < 0.05) than their N counterparts. Within N and I ranges, E steaks had lower TBARS (*p* < 0.05) and greater MRA (*p* < 0.05) than C steaks; however, I steaks presented similar results in MRA only (*p* < 0.05).

Neither pH_u_ ranges nor enhancement affected the MRA result over the 14 days of the experiment, except for the C-N steaks, which showed a decrease after day 5 (*p* < 0.05). Regarding lipid oxidation, enhancement promoted a remarkable stability in I and H steaks throughout MAP time, contrasted with the trend observed in C-I steaks (0 < 5 < 14, *p* < 0.05) and C-H steaks (0 < 5–14, *p* < 0.05). Within the N range, enhancement seemed to attenuate the progression of lipid oxidation, with a single growth between days 0 and 5 (*p* < 0.05, instead of the continuous increment showed by C steaks (0 < 5 < 14, *p* < 0.05). Surface MetMb accumulation also followed the pH-dependent pattern. Normal pH_u_ steaks had a progressing increase until day 8 (*p* < 0.05), whereas I samples only showed an increase between day 5 and day 8 (*p* < 0.05), and H samples exhibited stable values.

## 4. Discussion

### 4.1. Muscle pH and Headspace Composition

As the pH_u_ ranges increased, the O_2_ and CO_2_ concentrations varied more over the time, while the *Longissimus* muscle pH values showed more stability. These results agree with those reported in other studies [28,29], where there was an increase in pH values at the end of the experiment, regardless of the muscle or modified atmosphere used.

The pH changes during aerobic storage and display may also be a consequence of the microflora on the samples. Slight increases or decreases in pH during aerobic storage are not unusual, and the direction of pH change may be dependent on bacterial populations present on the samples [30].

The enhancement had no significant effect on the pH of M. *Longissimus lumborum* steaks samples, which was comparable to the pH of the control samples after all storage and during display time. Previous studies also reported no change in the beef cuts’ pH post-injection with similar levels of lactate [20,22,28]. The pH results of this study indicate that the pH differences established at the beginning and maintained during retail display after storage time are more related to the postmortem pH drop than to the brine enhancement.

The pH changes during aerobic storage and display may also be a consequence of the microflora on the samples. Slight increases in pH during aerobic storage are not unusual, and its extension may be dependent on the aerobic bacteria populations present in the samples, whose metabolites along with products of deamination of proteins, such ammonia [30]..

The time-related stability in muscle pH and changes in headspace gas composition exhibited by H steaks were reported previously [15,31]. An increase in pH stimulates myofiber aerobic metabolism and microorganism growth, resulting in the production of CO_2_ at the expense of O_2_ [32,33]. Furthermore, oxygen can bind to heme proteins, solubilize in the muscle, and participate in enzymatic and non-enzymatic oxidative reactions [34]; however, the latter might have been limited since these steaks exhibited great oxidative stability (Section 3.3).

### 4.2. Color Development and Stability of Antioxidant-Enhanced Steaks

The mitochondrial OC and MRA affect Mb redox formation and then the meat color development and stability. The improvement in OC and MRA observed in E-H steaks, as its instrumental darkening, have been reported by previous findings [9,14,35,36,37].

According to Bendall & Taylor [32], at pH 7.2, mitochondrial OC is 50–75% faster than at pH 5.8. The addition of lactate in beef, as demonstrated by Kim et al. [28], increases the concentration of reduced nicotinamide adenine dinucleotide (NADH), from endogenous NAD+, via lactate oxidation by mitochondrial lactate dehydrogenase (LDH). Enhancement then supplemented the lactate content from the postmortem glycolysis, which gradually declines during the storage [36]. 

The additional concentration of NADH prolongs the electron transfer in complex I of the mitochondrial membrane, thus ultimately boosting the reduction of oxygen to water [28,36]. NADH-triggered faster OC, therefore, would deplete the content of oxygen, resulting in a thicker surface layer of the dark and dull purplish-red DeoxyMb to the detriment of the bright cherry-red OxyMb [35,36]. The surface darkening of the steaks is then determined by the changes in the reflectance and absorbance properties, as observed on day 0 in the current study. The mitochondria-mediated darkening of meat was attested by Ramanathan et al. [38] after using the complex-I-inhibitor rotetone. This compound disrupts the mitochondrial electron transport chain (ETC) and, therefore, the oxygen consumption, which underlies the reversion on darkening in meat color found by the authors.

Once formed, MetMb may be reduced to DeoxyMb, which can be oxygenized into OxyMb by a pivotal system called MetMb-reducing ability, which comprises enzymatic, non-enzymatic, anaerobic medium, and electron-transfer-mediated mechanisms [34]. The enzymatic pathway involves the oxidation of NADH by the enzyme NADH-dependent cytochrome b5 (MetMb reductase) while MetMb is reduced [28]. Higher pH, in turn, enhances MRA for accelerating the activity of MetMb reductase until its maximum at pH 7.3, as demonstrated by Echevarne et al. [39]. Furthermore, MetMb can also be reduced after receiving electrons from complexes III and IV during the ETC in the inner mitochondrial membrane [34], in contrast to the stimulated production of pro-oxidants, such as reactive oxygen species (ROS) (in special superoxide radical and oxygen peroxide) in complex I [9,40]. In addition, lactate enhancement and/or the higher pH_u_ can also enhance other antioxidant biochemical pathways, thus extending color and oxidative stability [29,34].

Besides the aforementioned increase in OC and thus favorable DeoxyMb accumulation on the steak surface, Ramanathan et al. [21] showed high pH_u_ (6.4), and lactate-enhancement had tighter interfiber spaces than those of normal pH_u_ (5.6) and non-enhanced and water-enhanced, respectively. The higher muscle pH is recognized as being a darker color, greatly because of a lower light scattering and greater light absorption [11]. A closer structure also impairs the proper oxygen penetration, which is essential to produce a surface layer of OxyMb. The pH-induced darkening is also related to being the result of lactate- and phosphate-enhancement due to their alkalinity, which would raise the water-holding capacity (WHC) of muscle proteins and thus produce swollen muscle fibers [11]. Nevertheless, in the present experiment, brine injection (with an average of 2.25% lactate) did not increase muscle pH, which does not completely support our findings. Furthermore, lactate may also increase the refractive index of the sarcoplasm proteins, thereby decreasing light scattering [41].

Previous research also reported darker red color on Longissimus steaks with the combination of lactate and rosemary in HiOx-MAP [42]. These authors suggested that meat darkening may have resulted from a lactate-induced maintenance of surface DeoxyMb rather than to MetMb formation. Several authors have reported a darkening effect of lactate enhancement on the beef surface [28,29,42]. Besides the NADH-mediated darkening of meat, lactate may also have increased water holding capacity, reducing surface moisture and light scatter, which increased the surface dark color intensity [43].

These results are in contrast with other findings. For example, Knock et al. [20] did not observe any effect of the addition of potassium lactate on the values of L* values of the rib steaks. On the other hand, the use of calcium lactate in conjunction with phosphate increased the values of L* of beef round muscles in HiOx-MAP [30]. Lower L* observed in H pH_u_ is in agreement with what is reported by Abril et al. [44], McKeith et al. [8], and English et al. [31]. Dark cutting in high pH_u_ has been associated with a higher water-holding capacity of miofibrillar proteins, with decreasing in light scattering inside the structure of fiber as described by Hughes, Clarke, Li, Purslow, and Warner [11]. Darkening causes a substantial economic loss to the fresh meat industry [43]. The muscle may have pH_u_ higher than 5.8 due to numerous pre-slaughter stress factors reviewed by Ponnampalam et al. [5]. Pre-slaughter stress results in low muscle glycogen concentration and the production of lactic acid in the carcass, a defect known as dark-cutting beef [5]. Nellore bull and their crossbreeds, widely used in beef cattle production in several Latin American countries, such as Brazil, may have a greater susceptibility to stress than castrates and females and, therefore, darker in color [45], as can be observed in the H steaks samples of our study.

However, darkening could also have resulted from other compounds present in the brine solution. The tripolyphosphate-induced darkening can derive from structural changes due to an increase in the muscle pH [29] as well as in the expansion of myofibers and in extraction of myofibrillar proteins [46], although no difference in pH values after brine-enhancement was detected in the current research. Wills et al. [47], on the other hand, reported greater L* values by the use of 0.1% rosemary in LL beef steaks (pH_u_ > 6.0) packaged in PVC, indicating enhancement lightened those steaks. The authors associated such as a result of the water added by injections, which may have improved surface reflectance and Mb oxygenation, from oxygen carried by injections.

The surface-dark color of enhanced and non-enhanced steaks with pH_u_ > 5.8 was substantially upgraded by storage in HiOx-MAP for 5 days and no significant discoloration throughout display time, as also found in recent studies [14,15,47,48]. High oxygen partial pressure in HiOx-MAP provides sufficient oxygen to meet pH- or lactate-induced greater respiratory demands, as well as to diffuse deeper into the muscle and then bind to Mb thickening the surface layer of the bright and cherry-red OxyMb. In addition, chilling conditions during MAP storage may have contributed to color improvement because in as much as oxygen, solubility increases as temperature falls [40].

The aerobic packaging, however, reduced OC of non-enhanced greater-than-normal pH_u_ steak. As demonstrated previously [9,35,49], OC decreases during storage/display time primarily due to the depletion of substrates to maintain mitochondrial functionality and NADH production. Given the faster OC reported in steaks with pH_u_ > 5.8, a greater fall in OC after HiOx-MAP is expected, in relation to those of N pH_u_, thus resulting in a more intense depletion of NADH, for example. This hypothesis is consistent with the lower decrease shown by enhanced steaks pH_u_ > 5.8, which indicates that NADH was formed by the lactate injected, due to an extended metabolic activity, as found by Kim et al. [28].

Myofibers of H steaks have greater structural integrity, which limits the attack by reactive species. Proteomic findings of *Bos taurus* steaks with pH_u_ of 6.86 showed overexpression of large 70 kDa heat shock proteins (HSP70), such as GrpE protein homolog 1 (GRPEL1) and DnaJ homolog subfamily C member 11 (DNAJC11) [37]. Similarly, Poleti et al. [24] reported Nellore bull steaks with pH_u_ > 6.0 with overexpression of small HSP27, considered crucial for beef color [48], as well as more antioxidant proteins, such as zeta crystalline (CRYZ) and NAD(P) transhydrogenase mitochondrial (NTT). These overexpressed chaperones (HSP 27 and 70) maintain the integrity of membranes, repair structures, and inhibit apoptosis against stress-induced denaturation of myofibrillar and sarcoplasmic proteins [37,41,50]. Particularly, HSP27 also prevents the activation of caspase 3 by impeding the translocation to mitochondria of the pro-apoptotic protein Bid, which mediates the release of cytochrome c, another pro-apoptotic, thereby preventing apoptosis [50]. Finally, the oxidative stability of dark cuts may also be affected by the greater content of mitochondria in these muscles, which might reduce pyridine nucleotide to produce NADH, then used by MetMb reductase [8].

Furthermore, the oxidizing environment inside the trays can also induce the overexpression of antioxidant (peroxiredoxin 2 and 6) and chaperone (protein DJ-1 and HSPs) proteins to prevent structural damage in the mitochondria and then leakage of an oxidative enzyme, as well as balance the cell redox state [51]. The greater oxidative stability associated with the high oxygen concentration maintained the MAP-induced bright-red color in greater-than-normal pH_u_ steaks, despite time-related changes.

Enhanced steaks with pH_u_ > 5.8 synergistically combined both antioxidant protections then resulted in minor variation in lipid oxidation and MRA during 14 days of MAP time, despite the oxidant environment that steaks underwent (HiOx MAP and display under fluorescent lighting),in the present experiment. Wills et al. [47], contrarily, did not find a difference for TBARS between non-enhanced and rosemary-enhanced steaks with pH_u_ > 6.3 packaged in HiOx atmosphere (after 7 days aging). A hypothesis for the lack of results of the oxidative protection may be attributable to the individual use of 0.1% rosemary extract on dark cuts in opposition to the blend used in the present experiment, which may endorse the antioxidant efficacy of combining lactate and rosemary extract.

Contrastingly to the effect on I and H steaks, HiOx-MAP was not able to ameliorate the surface color of N steaks to the same extent. Potentially this result derived from the initial higher Mb oxygenation combined with a lower OC, which may have limited a significant additional OxyMb formation, whose accumulation may have led to the observed higher a*, b*, and chroma and smaller hue angle values. This hypothesis is supported by our data of OC and surface proportions of OxyMb and DeoxyMb, which were stable between days 0 and 5. Conversely, N steaks underwent a meaningful discoloration after fluorescent lighting (days 8), not exhibited by those of I and H ranges, indicating photo-oxidation.

Djenane et al. [23] demonstrated a parallel color deterioration after displaying C-N *Longissimus dorsi* packaged in HiOx MAP (70% O_2_) under fluorescent lighting (1000 lux) at 25 °C. Similar to our results, the authors also reported greater Mb and lipid oxidation, with consequent lower a*. The pro-oxidant effect of ultraviolet (UV) radiation from lamps along with an oxidizing atmosphere was attributed to have been causative since other lighting conditions (conventional fluorescent with UV filter, low-UV color balanced lamp, and darkness) did not promote such oxidation and discoloration.

Endogenously, lower pH, such as that exhibited by N steaks, can stimulate iron release from Mb and other iron-carrying molecules. Free iron, in turn, can act as pro-oxidants or interact with Mb to producing the also oxidant ferrylmyoglobin and ultimately amplify mitochondrial ROS production, such as HO∙ and H_2_O_2_, which can initiate or extend the oxidative chain reaction. In addition, postmortem proteolysis of muscle proteins reduces physical barriers, thus facilitating ROS access to susceptible molecules, such as the membrane’s PUFAs and during storage there is inactivation of some antioxidant enzymes [9,40,49,52,53].

As shown by Wu et al. [37], *Bos taurus*’ N steaks overexpressed myozenin-3 (MYOZ3) and myomesin (MYOM2), which enhance myofibril fragmentation and then enable oxidized products to amplify the reaction through lipid, protein and Mb oxidation [14,48]. This lower structural integrity, combined with a time-related reduction in the content of reducing equivalents and endogenous antioxidants, as well as in enzymatic antioxidant activities, minimizes the cell’s ability to reduce oxidized molecules or prevent further oxidations [9,49].

Regarding the substrates, grass-fed cattle have been reported to produce beef richer in PUFAs [3,4], especially omega-3 [54]. Although the daily intake of PUFAs and mostly omega-3 has been recommended due to their health benefits, these fatty acids are highly susceptible to being oxidized. To illustrate, omega-6 linoleic acid (C18:2) and omega-3 linolenic acid (C18:3) were found to be oxidized 10 and 20–30 times faster than oleic acid (C18:1), respectively [55].

Exposing steaks richer in PUFAs and inefficient to replenish or maintain endogenous antioxidants to pro-oxidant agents may have increased lipid oxidation, despite the low TBARS values exhibited in the present study. ROS, lipid free radicals, and secondary lipid oxidation products, such as 4-hydroxy-2-nonenal (HNE), can deteriorate meat color via MetMb formation [52]. Furthermore, HNE may also worsen color by its covalent binding to cysteine, histidine, lysine, and arginine residues [52,56] of oxidoreductase enzymes, such as LDH [56] and glutathione reductase [57], as well as Mbs histidine residues [52]; besides inhibit electron-transport mediated MRA [38] and accelerates the oxidation of the OxyMb already formed.

In contrast to C groups, E-N steaks showed a synergistic effect on MRA and TBARS (between days 5 and 14). This higher color and lipid stability supports the oxidative protection by antioxidant-enhancement solution used in this study, which scavenges ROS and then inhibits further oxidation, chelates free iron, and then limits the initiation step, respectively [19]. The sequestrant ability of tripolyphosphate is particularly helpful to prevent the iron-induced ROS formation in HiOx packages, thereby limiting the oxidation of macromolecules [18,19]. In agreement with our findings, Lu et al. [14] also found fewer TBARS values in C-H steaks than in those of C-N pH_u_, even in HiOx-MAP and fluorescent lighting, which reinforces the pH-dependent antioxidation.

These results demonstrate the application of antioxidant enhancement, and HiOx-MAP may be a tool to develop and stabilize the color of greater-than-normal pH_u_ steaks. Finally, given the molecular particularities reported to distinguish beef produced in Brazil, it is suggested a deeper investigation regarding novel techniques, such as genomic, proteomic, and metabolomics, to provide a better understanding of the relation molecular pre-harvest factors and the strategies to improve color development and stability of Nellore bull greater-than-normal pH_u_ steaks.

## 5. Conclusions

Nellore bull steaks of intermediate or high ultimate pH groups had more redness and color stability than those of normal pH_u_. On the other hand, high pH_u_ steaks enhanced with 2.5% potassium lactate/0.3% phosphate and 0.06% rosemary extract followed by HiOx-MAP had lower TBARS values than non-enhanced control steaks. However, the benefit of color stability offered by HiOx-MAP was reduced by the level of potassium lactate/phosphate used in this study as follows: enhanced steaks were darker (lowest L* values) than control treatment during the storage and display period. Perhaps the Nellore breed used in this study is less able to convert lactate to pyruvate via lactate dehydrogenase than animals of other breeds. Future work using animals of different breeds may be useful in elucidating the proteomic profile and biochemical processes that regulate color stability as a function of genetic variation.

## Figures and Tables

**Figure 1 foods-12-01302-f001:**
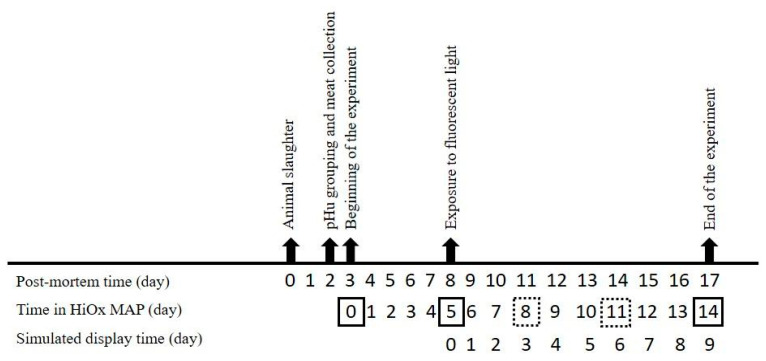
Experimental scheme of the study. HiOx-MAP: high oxygen modified atmosphere packaging (80% O_2_/20% CO_2_). Boxes with continuous line indicate the days when all measurements were performed. Boxes with dotted lines indicate the days when metmyoglobin reducing ability (MRA), oxygen consumption (OC) and lipid oxidation (TBARS) were not measured.

**Figure 2 foods-12-01302-f002:**
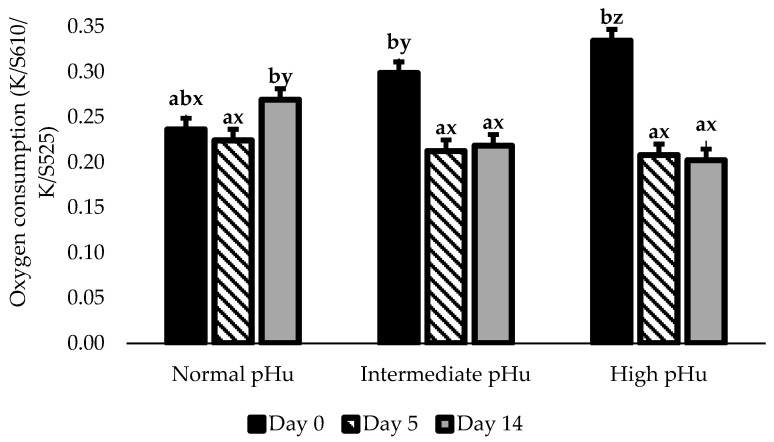
Oxygen consumption, measured as K/S 610/K/S 525) of Nellore *Longissimus lumborum* steaks with three ultimate pH (pH_u_) packaged in HiOx-MAP. ab Difference among days within a pH_u_ range (*p* < 0.01). xyz Difference among pH_u_ ranges within an experimental time (day) (*p* < 0.01). pH_u_ ranges: normal (*n* = 12; 5.40 < pH_u_ < 5.79), intermediate (*n* = 12; 5.80 < pH_u_ < 6.19) and high (*n* = 12; pH_u_ > 6.20). HiOx-MAP: high oxygen modified atmosphere packaging (80% O_2_/20% CO_2_). OxyMb: oxymyoglobin -lower value indicates greater OxyMb formed during air-blooming, and then lower oxygen consumption (OC). Standard error bars are indicated.

**Table 1 foods-12-01302-t001:** Brine composition and final residual concentration of each compound in enhanced muscles per ultimate pH (pH_u_) range.

Ingredients	Concentration in the Brine (%)	Final Residual Concentration of the Brine in the Enhanced Muscles (%) ^§^		
		Normal pH_u_ (*n* = 6)	Intermediate pH_u_ (*n* = 6)	High pH_u_ (*n* = 6)
Potassium L-lactate	27.50	2.16	2.29	2.31
Sodium chloride	3.30	0.26	0.27	0.28
Sodium tripolyphosphate	3.30	0.26	0.27	0.28
Rosemary oil extract	0.64	0.05	0.05	0.05
Cold mineral water	65.26	5.13	5.43	5.49
Total	100.00	7.86 ^†^	8.32 ^†^	8.41 ^†^

§ Obtained from the percentage of each ingredient in the brine solution and the injection level. † Injection level calculated as [(weight after injection − raw weight)]/raw weight]/100. pH_u_ ranges: normal (5.40 < pH_u_ < 5.79), intermediate (5.80 < pH_u_ < 6.19), and high (pH_u_ > 6.20).

**Table 2 foods-12-01302-t002:** Effects of ultimate pH (pH_u_) and storage and display time on meat pH and gas composition of Nellore *Longissimus lumborum* steaks packaged in HiOx-MAP.

Attribute	pH_u_	Time in HiOx-MAP (Day)					SE
		0	5	8	11	14	
pH	Normal	5.47 ^ax^	5.58 ^ax^	5.51 ^ax^	5.76 ^bx^	5.82 ^bx^	0.05
	Intermediate	5.83 ^aby^	5.87 ^abcy^	5.77 ^ay^	5.97 ^bcy^	6.00 ^cy^	
	High	6.28 ^abz^	6.36 ^abz^	6.28 ^az^	6.23 ^az^	6.41 ^bz^	
MAP O_2_	Normal	80.4 ^b^	80.1 ^b^	78.8 ^a^	80.2 ^by^	78.9 ^ay^	0.3
	Intermediate	81.1 ^d^	80.3 ^cd^	78.5 ^a^	79.8 ^bcy^	78.9 ^aby^	
	High	81.2 ^c^	79.7 ^b^	78.8 ^ab^	78.8 ^abx^	78.0 ^ax^	
MAP CO_2_	Normal	19.9 ^a^	20.7 ^b^	20.7 ^bx^	20.4 ^abx^	20.4 ^abx^	0.2
	Intermediate	20.0 ^a^	20.6 ^ab^	20.7 ^bx^	20.9 ^bx^	20.9 ^bx^	
	High	19.8 ^a^	20.7 ^b^	21.6 ^cy^	22.2 ^cy^	22.2 ^cy^	

^abcd^ Means with different letters in the same row are different (*p* < 0.01) for each attribute. ^xyz^ Means with different letters in the same column are different (*p* < 0.01) for each attribute. pH_u_ ranges: normal (*n* = 12; 5.40 < pH_u_ < 5.79), intermediate (*n* = 12; 5.80 < pH_u_ < 6.19) and high (*n* = 12; pH_u_ > 6.20). HiOx-MAP: high oxygen modified atmosphere packaging (80% O_2_/20% CO_2_). SE: standard error of the mean.

**Table 3 foods-12-01302-t003:** Effects of brine-enhancement on instrumental color parameters and surface myoglobin forms of Nellore *Longissimus lumborum* steaks packaged in HiOx-MAP.

	L* (Lightness)	a* (Redness)	b* (Yellowness)	Hue Angle	Chroma (Saturation)	DeoxyMb (K/S_473_/K/S_525_)	MetMb (K/S_572_/K/S_525_)	OxyMb (K/S_610_/K/S_525_)
Control	46.50 ^b^	24.81 ^b^	17.00 ^b^	34.30 ^b^	30.11 ^b^	0.94 ^b^	1.23	0.21 ^a^
Enhanced	43.28 ^a^	22.77 ^a^	15.25 ^a^	33.72 ^a^	27.42 ^a^	0.93 ^a^	1.23	0.30 ^b^
SE	0.263	0.261	0.211	0.179	0.324	0.002	0.004	0.005

^ab^ Means with different letters in the same column are different (*p* < 0.05). Control: non-enhanced steaks (*n* = 18); Enhanced: brine-enhanced steaks (2.5% potassium lactate, 0.3% sodium chloride, 0.3% sodium tripolyphosphate, and 0.06% rosemary extract) (*n* = 18). HiOx-MAP: high oxygen modified atmosphere packaging (80% O_2_/20% CO_2_). DeoxyMb: deoxymyoglobin; OxyMb: oxymyoglobin; MetMb: metmyoglobin—lower value indicates surface accumulation of each Mb state. SE: standard error of the mean.

**Table 4 foods-12-01302-t004:** Time-related changes at instrumental color parameters and surface myoglobin forms according to ultimate pH (pH_u_) ranges of Nellore *Longissimus lumborum* steaks packaged in HiOx-MAP.

Attribute	pH_u_	Time in HiOx-MAP (Day)					SE
		0	5	8	11	14	
a* (redness)	Normal	24.78 ^bz^	24.62 ^b^	21.17 ^ax^	20.70 ^ax^	21.65 ^ax^	0.74
	Intermediate	22.19 ^ay^	25.42 ^b^	24.36 ^by^	25.10 ^by^	25.85 ^by^	
	High	18.37 ^ax^	26.28 ^b^	25.13 ^by^	25.68 ^by^	25.60 ^by^	
b* (yellowness)	Normal	16.76 ^z^	16.75	15.25 ^x^	15.46 ^x^	15.86 ^x^	0.59
	Intermediate	14.43 ^ay^	17.61 ^b^	17.31 ^by^	17.32 ^by^	18.23 ^by^	
	High	10.47 ^ax^	17.21 ^b^	15.85 ^bxy^	17.02 ^bxy^	16.21 ^bx^	
Hue angle	Normal	33.91 ^ay^	34.15 ^axy^	35.64 ^by^	36.74 ^by^	36.13 ^by^	0.45
	Intermediate	32.86 ^ay^	34.66 ^by^	35.32 ^by^	34.61 ^bx^	35.12 ^by^	
	High	29.54 ^ax^	33.15 ^bx^	32.34 ^bx^	33.49 ^bx^	32.53 ^bx^	
Chroma (color saturation)	Normal	29.92 ^bz^	29.79 ^b^	26.11 ^ax^	25.85 ^ax^	26.85 ^ax^	0.91
	Intermediate	26.48 ^ay^	30.93 ^b^	29.89 ^by^	30.51 ^by^	31.65 ^by^	
	High	21.16 ^ax^	31.42 ^b^	29.74 ^by^	30.81 ^by^	30.37 ^by^	
DeoxyMb (K/S_473_/K/S_525_)	Normal	0.94 ^bz^	0.93 ^abx^	0.93 ^abx^	0.93 ^abx^	0.92 ^ax^	0.01
	Intermediate	0.92 ^ay^	0.94 ^bx^	0.94 ^bx^	0.94 ^by^	0.93 ^abx^	
	High	0.88 ^ax^	0.95 ^by^	0.96 ^by^	0.95 ^by^	0.94 ^by^	
OxyMb (K/S_610_/K/S_525_)	Normal	0.25 ^abx^	0.23 ^a^	0.27 ^bcy^	0.29 ^cy^	0.27 ^bcy^	0.01
	Intermediate	0.30 ^by^	0.22 ^a^	0.24 ^ax^	0.23 ^ax^	0.22 ^ax^	
	High	0.39 ^bz^	0.22 ^a^	0.25 ^axy^	0.23 ^ax^	0.23 ^ax^	
MetMb (K/S_572_/K/S_525)_	Normal	1.29 ^cy^	1.22 ^bx^	1.17 ^ax^	1.14 ^ax^	1.15 ^ax^	0.01
	Intermediate	1.28 ^cxy^	1.25 ^bcxy^	1.21 ^ay^	1.21 ^ay^	1.22 ^aby^	
	High	1.25 ^abx^	1.27 ^by^	1.26 ^abz^	1.26 ^abz^	1.24 ^ay^	

^abc^ Means with different letters in the same row are different (*p* < 0.05) for each attribute. ^xyz^ Means with different letters in the same column are different (*p* < 0.05) for each attribute. pH_u_ ranges: normal (*n* = 12; 5.40 < pH_u_ < 5.79), intermediate (*n* = 12; 5.80 < pH_u_ < 6.19) and high (*n* = 12; pH_u_ > 6.20). HiOx-MAP: high oxygen modified atmosphere packaging (80% O_2_/20% CO_2_). DeoxyMb: deoxymyoglobin; OxyMb: oxymyoglobin; MetMb: metmyoglobin—lower value indicates surface accumulation of each Mb state. SE: standard error of the mean.

**Table 5 foods-12-01302-t005:** Effects of treatment, ultimate pH (pH_u_) and time on MRA and lipid oxidation (TBARS) of Nellore *Longissimus lumborum* steaks packaged in HiOx-MAP.

Attribute	pH_u_	Group	Time in HiOx-MAP (Day)			SE
			0	5	14	
MRA (NO-MetMb, K/S_572_/K/S_525_)	Normal	Control	0.88 ^bw^	0.87 ^bw^	0.75 ^aw^	0.02
		Enhanced	0.89 ^aw^	0.89 ^aw^	0.89 ^ax^	
	Intermediate	Control	0.90 ^aw^	0.85 ^aw^	0.89 ^ax^	
		Enhanced	0.93 ^awx^	0.95 ^ax^	0.90 ^ax^	
	High	Control	0.96 ^ax^	0.95 ^ax^	1.01 ^ay^	
		Enhanced	0.98 ^ax^	1.01 ^ay^	1.01 ^ay^	
TBARS (mg malonaldehyde/kg steak)	Normal	Control	0.15 ^ay^	0.21 ^bz^	0.33 ^cy^	0.01
		Enhanced	0.10 ^awx^	0.18 ^byz^	0.22 ^bx^	
	Intermediate	Control	0.08 ^aw^	0.17 ^bxy^	0.22 ^cx^	
		Enhanced	0.12 ^axy^	0.12 ^aw^	0.12 ^aw^	
	High	Control	0.09 ^aw^	0.15 ^bwxy^	0.15 ^bw^	
		Enhanced	0.13 ^axy^	0.14 ^awx^	0.16 ^aw^	

^abc^ Means with different letters in the same row are different (*p* < 0.05) for each attribute. ^wxyz^ Means with different letters in the same column are different (*p* < 0.05) for each attribute. pH_u_ ranges: normal (*n* = 6; 5.40 < pH_u_ < 5.79), intermediate (*n* = 6; 5.80 < pH_u_ < 6.19) and high (*n* = 6; pH_u_ > 6.20). HiOx-MAP: high oxygen modified atmosphere packaging (80% O_2_/20% CO_2_). MetMb: metmyoglobin—Lower value indicates greater MetMb formed in the nitric oxide solution, and then lower metmyoglobin-reducing ability (MRA). TBARS: thiobarbituric acid reactive substances. SE: standard error of the mean.

## Data Availability

The data presented in this study are available on request from the corresponding author.

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
