# Peer review of "Improvement of Color and Oxidative Stabilities in Nellore Bull Dark Meat in High-Oxygen Package by Lactate and Rosemary Oil Extract"

_foods, 2023, doi:10.3390/foods12061302_

Round 1

Reviewer 1 Report

Foods

foods-2161124

Lactate and antioxidants improved color and oxidative stabilities in Nellore bull dark meat in high-oxygen package

Dear Editor,

The article deals with the determination of the effect of the injection of solution with lactate, phosphate and rosemary extract and packaging in high oxygen atmosphere on the color and oxidative stabilities of dark Nellore bull steaks. There is no novelty in the study and the aim is not clear! What is the relationship between the criteria studied? Microbiological analysis is essential in this type of research. The lack of microbial analysis is an important shortcoming. No information is given about the DFD meat problem. In addition, discussion sections should be improved! Finally, in the present study, there is no any new information.

Author Response

Dear Reviewer 1.

We thank you and the reviewers for your time and effort in reviewing our manuscript. The feedback has been invaluable in improving the content and presentation of the paper.

We have revised our manuscript according to all of your comments, as explained after the word ‘Answer’ in the point-by-point responses below.

Yours sincerely,

Prof. Carmen J. Contreras Castillo

Reviewer 2 Report

The idea of the article is good. More discussion is needed. Some points to improve the current format of the article will be mentioned below:

The title should be corrected. The title of the article should not be a sentence. It should be written in such a way that it is a statement about the study. My suggestion is: “Improvement of color and oxidative stabilities in Nellore bull dark meat in high-oxygen package by Lactate and rosemary oil”

Please choose keywords in such a way that they are not mentioned in the title in addition to helping to understand the concept of the research.

What was the basis for choosing the rosemary oil extract in this research?

Please mention in the introduction some sentences about lactate and the plant extracts used.

The purpose of the research should be well explained at the end of the introduction.

Regarding the use of plant extracts and the effect on color and antioxidant properties in meat, it is better to use similar articles. For example, you can use the article 10.1016/j.fbio.2022.101622.

Instead of providing statistics from Brazil, it is better to provide global statistics. Because this journal is international and not only for the audience of one country.

Statistical Analysis should be numbered under the second section.

Statistical analysis: In animal investigations, it is necessary to treat in such a way that all fixed and random effects are taken into account. The samples must be treated as one group per treatment, and the replication level must be clear. You need to include ALL the model terms. The authors could see the following article that may be helpful: "Biffin TE, et al. 2020. The effect of whole carcase medium voltage electrical stimulation, tenderstretching and longissimus infusion with actinidin on alpaca quality. Meat Sci. 164 Article 108107". It should be mentioned that all fixed and random effects need to be included for the analysis of all traits. Due to the experiment being replicated, thus this needs to be included as a random term in the model. You could also have samples within muscles as a random term. Please consider this confidently.

Doesn't the plant extract used have proteolytic properties? In these results, it seems that it is necessary to investigate the proteolytic activity of the extract. In addition, it is very necessary to perform the SDS-PAGE test to check this extract on the breakdown of meat proteins. You can take advantage of the many manuscripts that have been used to investigate the effect of plant extract on meat protein. I introduce some of these articles published in 2022 listed to you: 10.3389/fmicb.2022.846622; 10.22146/ijbiotech.66434; 10.1016/j.fbio.2022.101622; 10.3390/polym14153188

The discussion section needs more specific detailed comparative studies. This part is very inadequate. Please compare with similar works after presenting each result. Improve this section carefully.

Conclusion: what is the future of your findings? The conclusion is not insightful, what are your suggestions?

Minor editing of English language and style required.

Author Response

We thank you  the Reviewer2 for your time and effort in reviewing our manuscript. The feedback has been invaluable in improving the content and presentation of the paper.

We have revised our manuscript according to all of your comments, as explained after the word ‘Answer’ in the point-by-point responses below. Please see the attachment.

Yours sincerely,

Prof. Carmen J. Contreras Castillo

Round 2

Reviewer 1 Report

Dear Editor,

Once I looked at the reviewer response letter, I see that there is no any logical response to my comments. Therefore, I still have the same opinion about the paper. There is no novelty in the paper. I must reject it in one more time. 

Best regards

Reviewer 2 Report

Most of the corrections have been implemented.